# Cosmetic makeup enhances facial attractiveness and affective neural responses

**Tomohiro Arai** [1]*, **Hiroshi Nittono** [2]

1 Shiseido Co., Ltd. MIRAI Technology Institute, Kanagawa, Japan, 2 Graduate School of Human Sciences, Osaka University, Suita, Japan

* tomohiro.arai1@shiseido.com

**Data Availability Statement:** The data that support the findings of this study are available in FigShare with the identifier: https://doi.org/10.6084/m9.figshare.16755682.v1.

## Abstract

Although it is well established that cosmetic makeup enhances perceived facial attractiveness, few studies have examined whether facial makeup modulates neural responses to face images. This study investigated behavioral and attractiveness-related brain responses to self-applied makeup, focusing on the N170, early posterior negativity, P300, and late positive potential components of event-related brain potentials. A total of 77 Japanese women participated in two experiments (N = 34 and 43 for Experiments 1 and 2, respectively). Experiment 1 assessed the effects of self-applied makeup on attractiveness-related event-related potential amplitudes using facial images during a makeup identification task in which makeup was directly relevant to task demands. Experiment 2 examined the effects of self-applied makeup using images of one's own face and another female's face when performing a gender classification task, where the presence of makeup had no explicit connection to facial gender classification. In both experiments, faces with makeup were rated as more attractive and elicited more negative early posterior negativity and more positive late positive potential components, regardless of the participant's own face or another person's face. These findings suggest that people are spontaneously motivated to pay visual attention to faces with makeup, which supports the idea that makeup adds reward value to the facial appearance of the human. Moreover, neural evidence empirically confirmed that the benefits of makeup are not just limited to how others see your face but also extend to how you see your own face.

## Introduction

From ancient Greece to today, people, especially women, have used facial makeup to enhance their attractiveness [1]. According to the US Food and Drug Administration, cosmetics are defined as "articles intended to be rubbed, poured, sprinkled, or sprayed on, introduced into, or otherwise applied to human body. . . for cleaning, beautifying, promoting attractiveness, or altering the appearance" [2]. Today, several types of makeup products are used, sometimes to enhance facial contrasts to increase femininity and at other times to camouflage perceived skin deficits [3–5]. Facial painting has become increasingly fashionable and common among people, resulting in the growth of multi-billion-dollar cosmetic industries [1]. Many people purchase cosmetic products because they believe that the products modify their appearance [5].

**Funding:** This study was funded by Shiseido Co., Ltd. TA is an employee of Shiseido Co., Ltd. HN served as a member of a joint research project between Shiseido Co., Ltd and Osaka University, and did not receive any personal remuneration. The funder provided support in the form of salaries for TA as well as research funds, but did not have any additional role in the study design, data collection and analysis, decision to publish, or preparation of the manuscript. The specific roles of these authors are articulated in the 'author contributions' section.

**Competing interests:** The authors have read the journal's policy and have the following competing interests: TA is a paid employee of Shiseido Co., Ltd. HN serves as a member of a joint research project between Shiseido Co., Ltd and Osaka University. There are no patents, products in development or marketed products to declare. This does not alter our adherence to all the PLOS ONE policies on sharing data and materials.

## The effects of makeup on behavioral measures

Current research has attempted to empirically assess several effects of facial makeup. Morikawa and colleagues have used a psychophysical method and have revealed that observers perceived the eyes as larger in a photograph of a female model who applied eyeliner, mascara, or eye shadow compared to the photograph of a female model without any eye makeup, even though actual eye sizes were completely identical [6, 7]. These findings suggest that facial makeup alters the perception of facial features in the same way as geometrical visual illusions.

From the perspective of facial recognition, Ueda and Koyama [8] and Tagai et al. [9] found that facial makeup affects the judgment of recognizing the facial images of the same person. The facial images had three different styles of makeup: light (softer), heavy (more glamorous), and no makeup. Ueda and Koyama [8] reported that faces with light makeup were more accurately recognized than those without makeup, whereas heavy makeup decreased the accuracy of facial recognition. Contrastingly, Tagai et al. [9] reported that faces wearing light and heavy makeup had worse accuracy than those without makeup. Although their results were conflicting, their findings suggest that facial makeup alters the recognition of facial identity.

A series of previous studies have supported the positive effects of makeup on perceived facial attractiveness. Both male and female observers rated faces with makeup as more attractive than faces without makeup [10–18]. This appearance-enhancing effect has been repeatedly confirmed, both with self-applied makeup [10, 14, 16–18] and professionally applied makeup [9, 11–13, 15, 18]. Additionally, the Implicit Association Test (IAT) revealed that makeup use is subconsciously associated with positive evaluations [19].

Furthermore, the effects of makeup are not limited to the perceptions of others' faces and also affect the perception of one's own face [10, 16]. In a study reported by Cash et al. [10], women believed that facial makeup made them more attractive. Additionally, they showed that women tended to overestimate the attractiveness of their own faces with makeup compared to their peers' evaluations while underestimating those without makeup. Palumbo et al. [16] assigned female undergraduates to one of three manipulation groups: makeup, face-coloring, and music listening groups. The participants assigned to the makeup group were asked to apply makeup by themselves, those assigned to the face-coloring group were asked to color a schematic face, and those assigned to the music listening group were asked to listen to a music excerpt by Mozart. All participants were asked to rate their degree of self-perceived attractiveness before and after each manipulation. The authors found that only participants assigned to the makeup group reported an improvement in self-perceived attractiveness after the manipulation. The evidence we have reviewed thus far suggests that facial makeup modifies how faces look at both perceptual and cognitive stages, and the effects extend to how we see our own faces.

## The effects of facial makeup on neural measures

The idea that facial makeup can influence the perception and recognition of faces has also been supported by studies using neural measures. Ueno et al. [20] used functional magnetic resonance imaging (fMRI) to examine the effects of facial makeup. In their study, facial images with and without makeup were presented to participants during an fMRI scan, and participants were asked to rate the attractiveness of each face. They identified increased activation of the left medial orbitofrontal cortex when participants viewed faces with makeup compared to when they viewed faces without makeup. Their finding was in line with the findings of other studies, demonstrating that faces that were rated as being more attractive activated reward-related and emotion-related brain regions, including the ventral striatum and medial orbitofrontal cortex [21–27]. This empirical evidence supports previous findings that facial makeup adds a reward value to faces.

Recent studies employing the event-related potential (ERP) technique have illustrated the effects of makeup on facial perception [9, 28], particularly among early visual ERP components (P100 and N170). P100 is a positive deflection that peaks approximately between 80 and 110 ms after stimulus onset in the occipital region and has been assumed to be sensitive to low-level visual features in the face, such as local luminance and contrast differences or spatial frequency [29]. N170 is a negative component peaking approximately between 120 and 170 ms following stimulus onset in occipitotemporal electrode sites and is believed to reflect the structural encoding stage of the face during which the representations of holistic facial configuration are generated [30–32].

Tanaka [28] digitally edited facial images by adding red color to the lips to simulate lipstick or by adding blue color to the eyelids to simulate eyeshadow and compared the P100 and N170 amplitudes across three conditions: faces with lipstick, faces with eyeshadow, and unedited faces. The results showed more negative N170 amplitudes for faces with lipstick than for unedited faces; however, no significant differences in P100 amplitudes were identified across the conditions. The authors regarded the more negative N170 amplitudes observed for the lipstick condition as an indicator that the lipstick drew the observer's attentional resources to the mouth.

Tagai et al. [9] showed images of female models wearing three different makeup styles (light makeup, heavy makeup, and no makeup) to independent female observers and found that the light makeup condition elicited less negative N170 amplitudes than the heavy makeup condition. Moreover, the observers rated light makeup as the most attractive style among the three styles presented. They interpreted these results according to the processing fluency theory [33, 34], suggesting that the light makeup made an individual face look more similar to an average prototype. Although these two studies have different interpretations of the N170 effect, both suggest that facial makeup affects the early stage of face processing.

## ERP components related to facial attractiveness

Although previous ERP research on the effects of facial makeup focused on early visual ERP components, several studies have shown that not only early latency ERP components but also middle and late latency ERP components are involved in facial attractiveness. Among the early latency ERP components, several studies have identified N170 as being sensitive to facial attractiveness [34–36]. However, the literature shows inconsistent findings concerning how facial attractiveness modifies the N170 amplitudes, with some studies reporting more negative N170 amplitudes in response to more attractive faces [35, 36], while others reporting less negative ones [34].

In addition to N170, it has been reported that faces that scored higher in perceived attractiveness elicited middle and late latency ERP components, called early posterior negativity (EPN) and late positive complex (LPC) [35–42]. EPN is a negative deflection that peaks approximately 240–280 ms after stimulus onset in the occipitotemporal regions [43–45]. Following EPN (>300 ms after stimulus onset), the LPC, long-lasting positive potentials consisting of the P300, and late positive potential (LPP) manifested over the centroparietal regions [45–47]. These components are known to be highly sensitive to sexual and monetary rewards [44, 48], along with other emotion-loaded stimuli [49]. EPN has been proposed to index automatic attentional capture for grossly discriminating between affective and nonaffective stimuli [44, 50]. The P300 indexes the initial allocation of attentional resources to affective stimuli, whereas the LPP indexes sustained attentional engagement [51, 52]. Altogether, the EPN, P300, and LPP have been assumed to reflect increased attention to motivationally salient stimuli, and facial attractiveness can potentiate these ERP components. These findings, along

with several behavioral studies reporting that people are motivated to detect attractive faces automatically [53, 54] and to keep looking at attractive faces [21, 55], raise the issue of whether facial makeup affects not only the N170 waveform but also the EPN, P300, and LPP waveforms.

## The present study

The primary goal of this study was to explore the effects of makeup on ERP components elicited in response to facial images. Of particular interest were attractiveness-related ERP components ranging from N170 indexing configural processing of faces to EPN, P300, and LPP reflecting automatic or motivated attentional allocation to affective meanings of faces. Since facial makeup often links to an enhanced experience of attractiveness [10–18], faces with makeup are expected to elicit a change in the attractiveness-related neural activities. However, only two published studies have found the effect of makeup on N170 [9, 28]. Moreover, to our knowledge, no published research has clarified whether viewing faces with makeup evoke EPN, P300, and LPP. This is the first study conducted to answer these questions.

Furthermore, previous ERP research on facial makeup has focused on reactions to other individuals' faces, and no ERP studies have explored the effect of makeup on the response to one's own face. Previous research has found that makeup can not only modify how others see our facial appearance but also how we see our own faces [10, 16]. Additionally, self-perceived attractiveness enhancement through makeup was accompanied by improvement in self-reported mood [16]. These findings point to the importance of a more positive self-perception of physical appearance on well-being or life satisfaction [56]. Thus, the secondary goal of this study was to clarify whether makeup evokes greater attractiveness-related brain responses not only when presented with another person's face but also when presented with an individual's own face.

It is also important to determine whether the effects of makeup on ERP responses to faces are observed at an implicit level. In a prior study, IAT results suggested that makeup implicitly influences how individuals evaluate faces [19]. Furthermore, previous studies confirmed the effects of facial attractiveness or makeup on ERP responses even when the degree of attractiveness or the presence of makeup was not explicitly associated with what the task demanded participants to do [9, 28, 34, 35, 39–41]. There is a compelling reason to study the effects of makeup on attractiveness-related neural responses, regardless of which observer is clearly aware of the presence of makeup while performing a task.

Previous ERP studies used facial images of individuals with professionally applied makeup or using a digital makeup simulator [9, 28]. Although such procedures facilitate the control of the physical characteristics of makeup, it remains unclear whether we can extend the results of these studies to self-applied makeup in the general public, which underscores the necessity of evaluating the influences of normal makeup use. Indeed, Batras et al. [18] reported that professionally applied makeup was rated as less natural looking than self-applied makeup, despite the fact that the former was rated as more attractive than the latter, suggesting that self-applied makeup has higher ecological validity for non-makeup specialists.

Although fMRI studies have demonstrated that makeup activates emotion-related and reward-related brain regions [20], previous literature indicated that facial impressions, including facial attractiveness, were appraised in as little as tens to hundreds of milliseconds [57, 58], suggesting that the ERP techniques, which have an excellent temporal resolution, are more suitable for studying neural activities associated with rapidly unfolding cognitive and affective processes that can be modulated by facial makeup.

Based on this background, we conducted two experiments to address whether makeup elicited a change in ERP responses to faces. We examined the face-evoked ERP responses to an

individual's own face with and without makeup in a face classification task where participants needed to look carefully at facial images. We studied this issue by employing different face classification tasks in Experiment 1 and 2. Experiment 1 examined whether viewing one's own face with makeup changed the attractiveness-related ERP amplitudes in a task where makeup was directly related to the classification of faces, whereas makeup had no clear connection to face classification in Experiment 2. Experiment 2 also investigated whether the effects of makeup on attractiveness-related ERP amplitudes occurred both in response to one's own face and in responses to another person's face. In both Experiments 1 and 2, we asked participants to self-apply their facial makeup.

In addition to the attractiveness-related ERP components, P200 was also assessed. P200 is a positive deflection that peaks approximately 200–250 ms after stimulus onset over occipito-temporal sites and has been recently reported to be a more valid and reliable neural index of self-face processing [59–61]. A series of experiments by Alzueta et al. reported that one's own face reduced the P200 amplitudes as compared to other individuals' faces [60, 61]. Because this study included discrimination between an individual's own face and another person's face, we confirmed whether makeup affects the ERP component that is known to be sensitive to self–other face discrimination.

Three hypotheses were tested in this study. First, we hypothesized that self-applied makeup enhanced perceived facial attractiveness. Second, we hypothesized that makeup led to a change in attractiveness-related ERP amplitudes, both with an individual's own face and with another person's face. It was expected that faces with makeup would elicit a larger negative deflection in the EPN time window and a larger positive deflection in the P300/LPP time window than faces without makeup. However, the mechanism by which facial attractiveness modifies the N170 amplitudes remains controversial. Nevertheless, relatively more findings supported more negative N170 amplitudes in response to facial attractiveness [35, 36], prompting the authors to conclude that makeup on faces would lead to more negative N170 amplitudes. Third, we hypothesized that the effects of a facial makeup on attractiveness-related ERP amplitudes would be independent of the makeup's task relevance.

## Experiment 1

### Method

**Participants.** Thirty-four Japanese women participated in this experiment (mean age = 30.03 years, standard deviation (SD) = 2.87 years). They were recruited from the general public by a Japanese marketing company and were paid JPY 7,000 (approximately USD 67) for their participation. Prior to the experiment, an online survey was conducted to identify potential participants. The eligibility requirements for participating in the experiment were as follows: (a) aged between 25 and 35 years; (b) wearing facial makeup for at least five days each week; (c) having a normal or corrected-to-normal vision; (d) having normal color vision acuity; (e) no current neurological or psychiatric disorders; (f) no history of neurological or psychiatric disorders; (g) no shrapnel or other metal or electronic implants in the body; (h) no skin disorders; (i) no scars or tattoos on the face; (j) no eyelash extensions within three months prior to the experiment; and (k) right-handedness, as determined by the Flinders Handedness survey (FLANDERS) questionnaire score between +5 and +10 [62]. Additionally, the respondents were asked to indicate the extent to which they agreed that their own facial makeup suits them on a scale ranging from 0 (completely disagree) to 10 (completely agree). To prevent a situation where every participant had a very strong belief that their own facial makeup makes them more attractive, we recruited 17 participants from among the respondents who scored >5 and 17 participants who scored <5.

We recruited only women because they were expected to self-apply their facial makeup and be able to distinguish between faces with makeup and without makeup much easier than men, which was assumed to be necessary for participants to complete our experimental procedure. Previous studies have revealed that perceived facial attractiveness correlated with the actual age of subjects who were photographed [63]. Moreover, makeup modulated the perceived ages of the subjects differently depending on the subject's age [64]. Based on these studies, we regarded large age differences in participants as potential confounding variables that could not be ignored in our study. To ensure that the participants were comparable in terms of age, we limited participants to those aged between 25 and 35 years.

**Stimuli.** Participants viewed six color images of their own faces with and without facial makeup using three different head orientations (Fig 1A). Photographs were taken using a digital single-lens reflex (DSLR) camera (Canon EOS 5D Mark II, Canon Inc., Ota, Tokyo, Japan), with a strobe light in a portable photo studio box featuring light-emitting diode (LED) lights. The distance between the camera and the participant was 76.2 cm. The photo studio box had a white background. Each participant wore a black hair turban and a black cape before the photoshoot to hide their hair, ears, and clothes. They were photographed while assuming emotionally neutral expressions with the head facing front, left at a 45° angle, and right at a 45° angle. The original image size was 3,744 × 5,616 pixels. The colors of the images were corrected using the X-Rite Color Checker Passport (X-Rite Inc., Grand Rapids, MI, USA) and Adobe Lightroom (Adobe Inc., San Jose, CA, USA). These images were cropped to 2,000 × 2,000 pixels, focusing on the faces. The images were saved in mirror-reversed orientations at 400 × 400 pixels.

**Procedure.** The participants were asked to arrive at the test room with the facial makeup that they would normally wear while going out with their friends on the weekend, including foundation, eye shadow, eyebrow color, cheek color, and lipstick or lip gloss. Additionally, participants were asked to avoid consuming caffeine 6 h before the experiment, avoid smoking 3 h before the experiment, and avoid alcohol on the day or before the day of the experiment.

After greeting participants and obtaining written informed consent, the experimenters took portrait photographs of the participants with facial makeup using three different head orientations. Next, the participants completely removed their makeup using cleansers and were photographed again using the same three head orientations.

After the photography session, participants sat in a dimly lit and sound-attenuated room at a distance of 60 cm from the computer screen. A chin rest was used to maintain the viewing distance and maintain a steady head position. The experiment consisted of a makeup identification task, with continuous EEG recording, and a face rating task, which is described in detail below. The total duration of the experiment was approximately 90 minutes.

The stimuli were displayed at the center of a 27-inch LCD monitor (Acer Inc., Xizhi, New Taipei, Taiwan) against an identical light gray background color [RGB (198, 198, 198)]. Each facial image subtended a visual angle of 11.89° horizontally and 11.89° vertically.

During the makeup identification task, participants used button presses to classify six facial images as faces with makeup or without makeup as quickly and accurately as possible. Participants indicated their responses by pressing the left and right buttons on a Cedrus response pad (Cedrus RB-530, Cedrus Corporation, San Pedro, CA, USA) using the index fingers of both hands. The assignment of the response buttons (left/right–makeup/no makeup) was counterbalanced across participants. The task was divided into six trial blocks consisting of 40 face presentations per block, totaling 240 trials overall. Each picture was presented 40 times. Each trial started with a white square frame presented for jittered inter-stimulus intervals (ISIs), followed by the presentation of facial images for 1,000 ms (Fig 1B). ISIs of 1,400, 1,454, 1,508, 1,562, or 1,616 ms were selected. During each transition between makeup conditions, ISIs

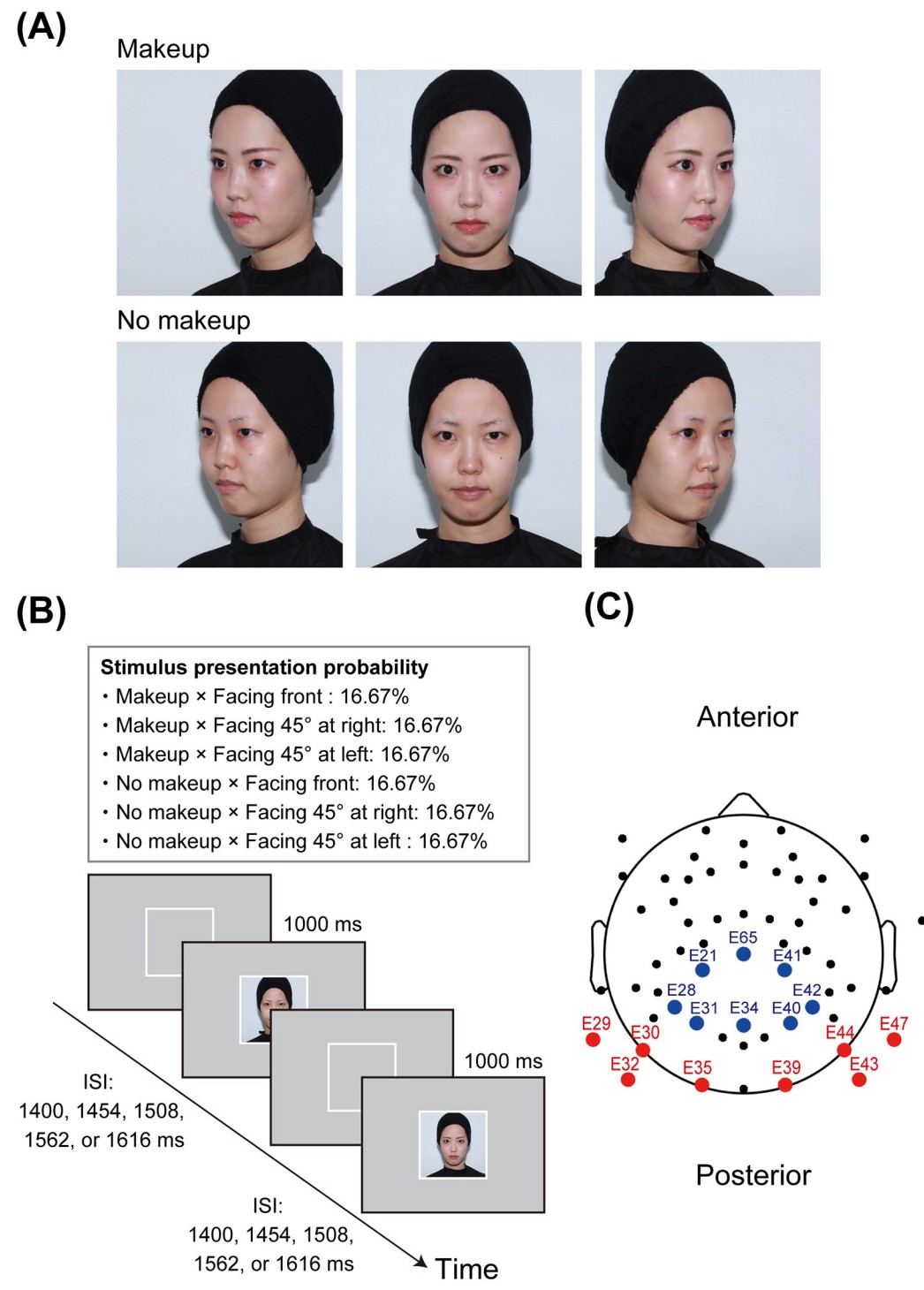

**Fig 1. Stimuli and procedures in Experiment 1.** Panel A depicts an example of facial images. The individual in this figure has given written informed consent to publish her facial images. Panel B illustrates a schematic representation of the makeup identification task. Panel C shows the EEG sensor layout and ROIs. The electrodes included in the occipitotemporal ROI are marked in red, and the blue circles indicate electrodes located in the centroparietal ROI. EEG, electroencephalogram; ROI, region of interest.

were regulated to be balanced (i.e., the possible conditional transitions were from makeup to no makeup, makeup to makeup, no makeup to makeup, and no makeup to no makeup). Therefore, facial images were presented in a pseudo-random order to ensure that the ISIs were balanced. The stimulus sequence was counterbalanced across participants. Before starting the task, participants completed 15 practice trials. The task was run using the Inquisit 4.0 software (Millisecond Software, Seattle, USA).

After completing the makeup identification task, the EEG scalp cap was removed and participants performed the face rating task. During this task, participants rated each individual image for facial attractiveness and 14 additional facial attributes using an 11-point Likert scale (0: completely disagree, 10: completely agree). The trial started with a white square frame, presented for 1,000 ms, followed by a facial image that was displayed until the participant responded. The facial attributes rated for each image are presented immediately above the image. Participants rated the six images in succession for each facial attribute before proceeding to the next facial attribute. The presentation orders of the facial images and facial attributes were completely randomized. The task was run using PsychoPy software [65].

All procedures were reviewed and approved by the ethical committee of the Shiseido Global Innovation Center. Data were collected prior to the COVID-19 pandemic outbreak.

**Electroencephalogram recording and analysis and ERP quantification.**   EEG data were recorded with a 64-channel Geodesics Sensor Net using Net Station 5.4 software (Electrical Geodesics Inc., Eugene, OR, USA). Amplified EEG signals were digitized using an A/D converter at a 1,000 Hz sampling rate with a 24-bit resolution. Continuous EEG data were recorded with respect to the Cz reference. The impedances were adjusted to below 50 kΩ, following the manufacturer's recommendations.

EEG data were analyzed offline using the EEGLAB toolbox, version 14.1.2b [66] running on MATLAB 2018b (The MathWorks, Inc., Natick, MA, USA). Four eye channels (E61, E62, E63, and E64) were not included in the analysis, resulting in a total of 60 electrodes. Zero-phase Hamming-windowed sinc finite impulse response filters were applied to continuous EEG data using the function pop_eegfiltnew(). The data were high-pass filtered with a lower passband edge of 0.1 Hz [transition band width = 0.1 Hz, cut-off frequency (−6 dB) = 0.05 Hz] and were low-pass filtered with a higher passband edge of 30 Hz [transition band width = 7.5 Hz, cut-off frequency (−6 dB) = 33.75 Hz]. The filtered data were downsampled to 256 Hz to reduce the computation time. Bad channels were identified using the TrimOutlier() plugin [67] and visual inspection. Signals from bad channels were discarded and then interpolated using an EEGLAB-based spherical spline interpolation algorithm. The EEG data were then re-referenced to the average of all 60 channels while adding the Cz channel back to the data set. Independent component analysis (ICA) using the extended informax algorithm was performed to suppress ocular artifacts, such as eye blinks and saccadic eye movements. The ICLabel plugin [68] and icaeyeblinkmetrics() plugin [69] automatically detected ocular components. We used E5 and E10 (corresponding to Fp1 and Fp2, respectively, in the international 10–20 system for EEG electrode placement) as dedicated artifact channels in the icaeyeblinkmetrics() implementation. After all independent components were visually inspected using the Viewprops() plugin [70], the ocular components were removed, and the remaining components were back-projected onto the scalp electrodes to obtain ocular artifact-free EEG data.

Preprocessed EEG data were segmented into epochs, beginning at 200 ms prior to stimulus onset and continuing for 1,000 ms after stimulus presentation. For each trial, waveforms were baseline corrected by subtracting the mean amplitude during the time interval 200 ms prior to stimulus onset from each data point in the waveform. Trials corresponding to incorrect responses were excluded from the analysis. Epochs were discarded if the signal amplitude variations exceeded −200 to 200 µV or the joint log probabilities deviated by six SDs from the

mean of the probability distribution for all trials at the single-channel level or three SDs at the all-channel level [71].

To quantify N170, P200, EPN, P300, and LPP amplitudes, the time windows and two regions of interest (ROI) were chosen a priori based on previous research [9, 28–32, 34–50, 59–61]. For the occipitotemporal ROI, we selected eight electrodes (E29, E30, E32, E35, E39, E43, E44, and E47) and determined a time window of 120–170 ms following the stimulus onset for N170 quantification, that of 200–230 ms following stimulus onset for P200 quantification, and that of 240–280 ms after stimulus onset for EPN quantification. For the centroparietal ROI, we selected eight electrodes (E21, E28, E31, E34, E40, E41, E42, and E65) and determined a time window of 300–500 ms following the stimulus onset for P300 quantification and that of 500–1,000 ms after stimulus onset for LPP quantification. The layout of the selected electrodes is presented in Fig 1C.

**Statistical analyses.** We performed a paired $t$-test to determine whether the differences between the two conditions (makeup vs. no makeup) for the same participant were not equal to zero. Three head orientations were combined in the analysis. The significance level ($\alpha$) was set to 0.05 for all analyses. We used a two-tailed test and computed the effect size (Cohen's within-subject $dz$) for the $t$-test. The dependent variables were: (a) attractiveness rating scores during the face rating task, (b) response accuracy during the makeup identification task, (c) median reaction times (RTs) associated with correct classifications during the makeup identification task, (d) mean N170 amplitudes, (e) mean P200 amplitudes, (f) mean EPN amplitudes, (g) mean P300 amplitudes, and (h) mean LPP amplitudes. The results for the other 14 attributes examined during the face rating task were not reported in this study. All statistical analyses were performed using R version 3.6.2 [72].

## Results

**Behavior.** Table 1 represents the summary of behavioral measures. For accuracy, no significant difference between the two conditions was identified for the makeup vs. no makeup classification accuracy ($t(33) < 0.001$, $p = 1.00$, $dz < 0.001$). For RT, no significant difference was observed between the two conditions in the makeup vs. no makeup classification RTs ($t(33) = 1.97$, $p = 0.057$, $dz = 0.338$), although the RTs were numerically longer for faces with makeup than those for faces without makeup. A paired $t$-test found a significant difference in attractiveness ratings between the two conditions ($t(33) = 8.71$, $p < 0.001$, $dz = 1.494$). Participants rated their own faces wearing makeup as more attractive than their own faces without makeup.

**Event-related potentials.** During the preprocessing of EEG data, an average of 0.09 channels (SD = 0.29, range: 0–1) was rejected, and an average of 2.38 independent components (SD = 0.65, range: 1–3) was removed as ocular components. None of the 16 channels located in the two ROIs were rejected by any participant. After rejecting bad epochs, not <86% of the original trials remained for ERP averaging per participant in each condition (makeup: M = 111.59 trials, SD = 3.62 trials; no makeup: M = 111.00 trials, SD = 3.53 trials).

**Table 1. Descriptive statistics for behavioral measures in Experiment 1.**

|  | accuracy[1] | reaction time[2] | attractiveness rating |
|---|---|---|---|
| makeup | 0.986 (0.017) | 580.29 (76.41) | 4.89 (2.32) |
| no makeup | 0.986 (0.020) | 567.63 (61.26) | 1.75 (1.54) |

Descriptive statistics are presented as mean (SD). RT, Reaction Time.

[1]Accuracy scores are expressed as the proportion of correct trials to total trials.

[2]Median RT per participant in each condition was calculated. The unit for the RT is milliseconds.

Fig 2 presents the grand average ERP waveforms over eight occipitotemporal and six centroparietal electrodes that were selected a priori for each condition (panel A), and the mean amplitudes of the N170, EPN, P300, and LPP waveforms for each condition (panel B). For N170, no significant difference was identified in the amplitudes between the two conditions ($t$(33) = 0.31, $p$ = 0.759, $dz$ = 0.053). For P200, no significant difference was identified in the amplitudes between the two conditions ($t$(33) = 1.87, $p$ = 0.071, $dz$ = 0.320). For EPN, a paired $t$-test found a significant difference in the amplitudes between the two conditions ($t$(33) = 2.71, $p$ = 0.011, $dz$ = 0.465). Viewing one's own face with makeup on resulted in more negative EPN amplitudes than viewing one's own face without makeup (makeup: M = 2.60 μV, SD = 2.62 μV; no makeup: M = 2.92 μV, SD = 2.44 μV). Note that more negative EPN amplitudes indicate more pronounced neural responses as the EPN is a negative-going deflection in the ERP waveform. For P300, no significant differences in the amplitudes between the two conditions were identified ($t$(33) = 1.00, $p$ = 0.323, $dz$ = 0.172). For LPP, a paired $t$-test found a significant difference in the amplitudes between the two conditions ($t$(33) = 2.83, $p$ = 0.008, $dz$ = 0.486). Viewing one's own face with makeup on elicited more positive LPP amplitudes than viewing one's own face without makeup (makeup: M = 1.42 μV, SD = 1.12 μV; no makeup: M = 1.06 μV, SD = 0.95 μV).

## Discussion

The results from the attractiveness rating replicated previous findings that makeup improved the perceived facial attractiveness of one's own face [10, 16]. The most important finding was that viewing one's own face with makeup elicited more negative EPN and more positive LPP waveforms than viewing one's own face without makeup. This finding was consistent with the assertion that attractive faces evoke more negative EPN and more positive LPP amplitudes than unattractive faces [35–42]. It has been reported that the EPN component signals the automatic attention capture of affective stimuli, whereas the LPP component indexes sustained attentional engagement with affective stimuli [44, 50–52]. Thus, we would argue that facial attractiveness enhanced by makeup draws an observer's attention involuntarily and motivates them to hold their attention. Another finding was that self-applied makeup, applied non-professionally by participants, was sufficient to evoke more pronounced EPN and LPP responses.

An unexpected finding was that the presence of facial makeup did not modulate the N170 and P300 amplitudes. This finding of the N170 waveform may be explained in part by individual differences in the makeup of participants. Tagai et al. [9] showed that the N170 amplitude was less negative for faces with light makeup than for those with heavy makeup and without makeup. In our study, we asked participants to self-apply facial makeup and did not strictly control the makeup style, resulting in a situation where some participants had heavier makeup and others had lighter makeup. This situation might have led to considerable variance in the effects of makeup on N170 amplitudes. One possible explanation for the lack of statistically significant differences in the P300 amplitudes is that identifying faces with makeup requires elaborate processing of incoming visual information. The median RTs were faster in response to an individual's own face without makeup than in response to an individual's own face with makeup, even though this trend did not reach the statistical significance. This indicated that the participants took longer to classify their own face with makeup, possibly because accumulation and elaboration of makeup-related information was necessary to ensure that the presented facial image had makeup. This may explain why we found no significant effect of makeup on the P300 component, which has been assumed to reflect the initial stage allocation of attention to motivationally salient stimuli [51, 52].

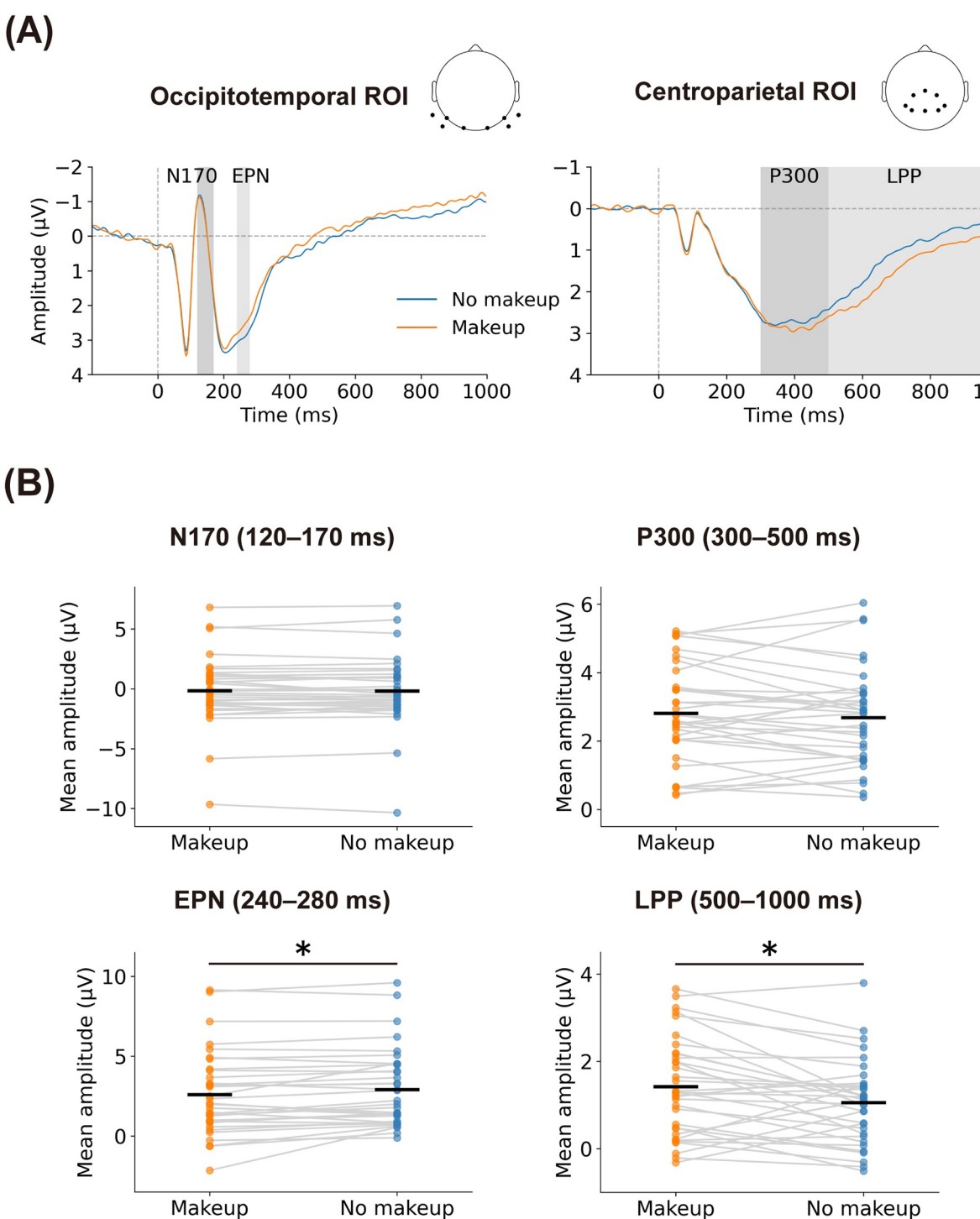

**Fig 2. ERP results in Experiment 1.** Panel A shows the grand average ERP waveforms elicited by facial images in the occipitotemporal ROI and centroparietal ROI. Panel B shows repeated measures plots visualizing amplitudes of the N170, EPN, P300, and LPP components for the conditions (makeup vs. no makeup). Horizontal black lines represent the mean amplitudes for each condition. Asterisks indicate a significant difference between the conditions determined by a paired $t$-test ($p < 0.05$). ERP, event-related potential; ROI, region of interest; EPN, early posterior negativity; LPP, late positive potential.

## Experiment 2

In Experiment 1, we observed the modulation of the EPN and LPP amplitudes in response to facial makeup. However, we did not include the images of other people's faces in this experiment. Moreover, whether makeup can modulate ERP amplitudes when it is implicitly associated with task demands remains unclear. The purpose of Experiment 2 was to test whether the more pronounced EPN and LPP activities induced by viewing faces with makeup were limited to one's own face or could be extended to the faces of other individuals during a gender classification task in which the presence of makeup had no explicit connection to facial gender classification.

### Method

**Participants.**   Fifty-four Japanese women participated in the study. They were recruited from the general public by a different Japanese marketing company and were paid JPY 14,000 (approximately USD 133) for their participation. Monitors were recruited through phone screening interviews. The eligibility requirements for participation in this experiment were the same as those in Experiment 1. Nine participants declined to participate in ERP measurement because of their poor health conditions, work circumstances, and pregnancy. Two were excluded from the analysis because of excessive sweat and movement artifacts. The final sample consisted of 43 participants (mean age = 30.74 years, SD = 3.18 years).

**Stimuli.**   Participants viewed five color images, consisting of two images of their own face (one with and one without facial makeup), two images of another female's face (one with and one without facial makeup), and one with a male's face (Fig 3A). A professional photographer took photos using a DSLR camera (Canon EOS 5D Mark II, Canon Inc.) with three strobe lights in a photographic studio. A photograph was taken against a black background, placing one strobe light on each side of the participant and one above the participant. The distance between the camera and the participant was 129 cm. Each participant wore a gray-colored hair turban and gray t-shirt before the photoshoot to hide their hair and ears and to control light reflection from clothes. They were photographed while assuming an emotionally neutral expression, with the head facing the front. The original image size was 3,744 × 5,616 pixels. The professional photographer manually obscured the hair, neck, ears, and clothes from the image using the brush tool in Photoshop CS6 (Adobe Inc.), leaving only facial features and contours. The colors of the images were corrected using the Photoshop software. The photographer carefully aligned the face size and eye locations between the two images of the same woman, one with makeup and the other without makeup. The images were saved in mirror-reversed orientations at 368 × 500 pixels.

Prior to the ERP experiment, the facial attractiveness and femininity of each participant without makeup were assessed in 200 independent samples (100 women; mean age = 29.95 years, SD = 3.03 years; age range: 25–35 years). Ten facial images of men were also included in the survey. We sorted participants who agreed to participate in the ERP experiment according to the obtained rating scores and paired images with similar attractiveness ratings, resulting in a total of 23 pairs. In each pair, two images were of the participant's own face, and the other two images were of another female's face. Ten facial images of men were sorted according to their scores. Each facial image of men was assigned to three pairs in ascending order. These pairings were performed to minimize the impact of differences in attractiveness levels between their own face and the faces of others that were presented.

**Procedure.**   Experiment 2 was divided into a photography session and an ERP experimental session. Each session was performed on a separate day. During the photography session, after obtaining written informed consent from participants, participants removed their

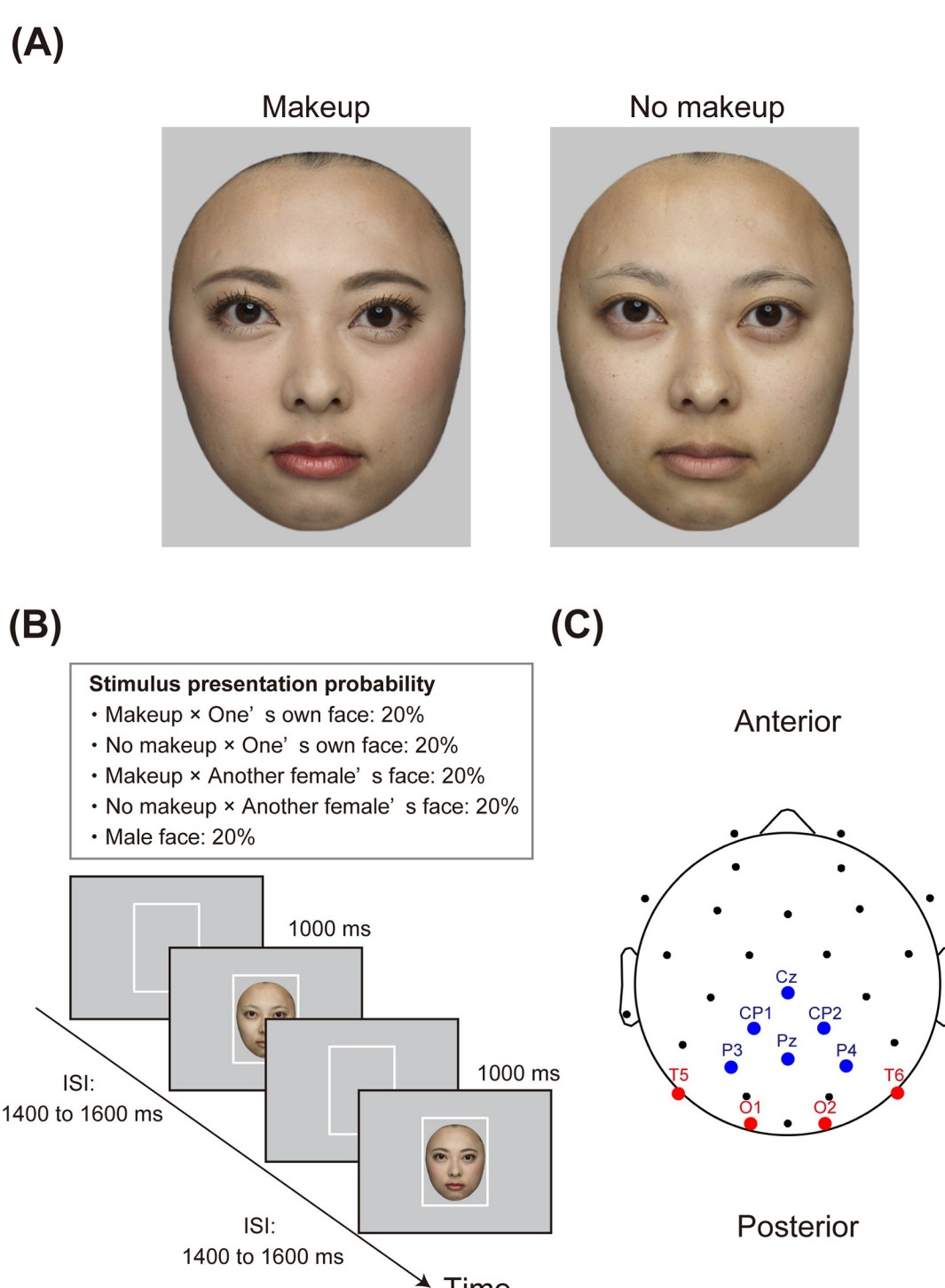

**Fig 3. Stimuli and procedures in Experiment 2.** Panel A depicts an example of facial images. The individual in this figure has given written informed consent to publish her facial images. Panel B illustrates a schematic representation of the gender classification task. Panel C shows the EEG sensor layout and ROIs. The electrodes included in the occipitotemporal ROI are marked in red, and the blue circles indicate electrodes located in the centroparietal ROI. EEG, electroencephalogram; ROI, region of interest.

makeup completely using cleansers and changed their clothes to a gray t-shirt. Then, a professional photographer took portrait-style photographs of participants without facial makeup. Participants were then asked to self-apply their makeup using the makeup style they would prefer while going out with their friends on a weekend, including foundation, eye shadow, eyebrow color, cheek color, and lipstick or lip gloss. After applying their makeup, they were photographed again using the same style as the previous one (without makeup).

After approximately four months (M = 132.40 days, SD = 4.21 days) following the photography session, participants were invited to participate in the ERP experimental session. The preparation protocols for participation in the EEG experiment were the same as those described in Experiment 1. Participants sat in a dimly lit room at a distance of 70 cm from the computer screen. A chin rest was used to maintain the viewing distance and prevent head movement. The experiment consisted of a gender classification task under continuous EEG recordings and a face rating task, as described in detail below. The total duration of the experiment was 120 minutes.

The stimuli were displayed at the center of a 22.5-inch LCD VIEWPixx Monitor (VPixx Technologies, Quebec, Canada) against an identical light gray background color [RGB (198, 198, 198)]. Each facial image subtended a visual angle of 8˚ horizontally and 10˚ vertically.

During the gender classification task, participants were asked to classify five facial images as either female or male via button press as quickly and accurately as possible. Participants pressed either the left or right button on a Cedrus response pad (Cedrus RB-530, Cedrus Corporation) with the index fingers of both hands. The assignment of response buttons (left/right–female/male) was counterbalanced across participants. The task was divided into 12 trial blocks consisting of 50 face presentations per block for a total of 600 trials overall, with each picture presenting for a total of 120 trials. Each trial started with a white rectangular frame presented for different ISIs, followed by the presentation of a facial image for 1,000 ms (Fig 3B). The ISIs ranged from 1,400–1,600 ms. Facial images were presented in a random order. Before starting the task, each participant completed 15 practice trials. The task was run using the Inquisit 4.0 software (Millisecond Software).

After completing the gender classification task, the EEG scalp cap was removed, and participants continued the face rating task. We did not include male facial stimuli during the task. The procedures for this task were the same as those described for Experiment 1, except that the participants rated their perceived emotions in response to each stimulus using the Self-Assessment Manikin (SAM) scale [73] at the end of the task. Perceived emotions were assessed in terms of emotional valence (on a scale ranging from 1 = unpleasant to 9 = pleasant) and arousal (on a scale ranging from 1 = calm to 9 = aroused).

All procedures were reviewed and approved by the ethics committee of the Shiseido Global Innovation Center. Data were collected prior to the COVID-19 pandemic outbreak.

**Electroencephalogram recording and analysis and ERP quantification.** The EEG data were recorded using an elastic cap and an ActiveTwo EEG acquisition system (Biosemi, Amsterdam, The Netherlands). A total of 32 Ag-AgCl active electrodes (FP1, FP2, AF3, AF4, F7, F3, Fz, F4, F8, FC5, FC1, FC2, FC6, T7, C3, Cz, C4, T8, CP5, CP1, CP2, CP6, T5, P3, Pz, P4, T6, PO3, PO4, O1, Oz, and O2) were used according to the international 10–20 system. Two additional external electrodes were placed on the left and right mastoids. The amplified EEG signals were digitized by an A/D converter at a 2,048 Hz sampling rate with a 24-bit resolution. The voltage from each active electrode was measured online with respect to a common mode sense active electrode, producing a monopolar (non-differential) channel. The electrode offset was maintained at less than ± 40 mV. We used a different EEG recording system than the one that was used in Experiment 1 because of laboratory constraints. However, both systems have been reliably used in many previous studies.

The EEG preprocessing procedure and ERP quantification methods were the same as those described for Experiment 1, except for the predefined electrode sites. Here, we selected four electrodes (T5, T6, O1, and O2) for the occipitotemporal ROI, and six electrodes (Cz, CP1, CP2, Pz, P3, and P4) for the centroparietal ROI. The layout of the selected electrodes is presented in Fig 3C. Note that the EEG data were re-referenced to the average of all 32 channels.

**Statistical analyses.** We performed a 2 × 2 repeated measures analysis of variance (ANOVA) on each dependent variable, with the facial identity (one's own vs. another female's face) and makeup (makeup vs. no makeup) conditions as within-subject factors. The male face condition was not included in the analysis. The significance level ($\alpha$) was set to 0.05, and we calculated the $\eta_G^2$ effect sizes for all ANOVAs. Additionally, in order to facilitate comparisons between Experiments 1 and 2, the main effect for the makeup factor was assessed by calculating the mean and SDs averaged across self-face and another female's face, and then its effect size was shown using Cohen's within-subject *dz*. The dependent variables were the same eight variables described for Experiment 1 plus the SAM valence and arousal rating scores for a total of 10 variables.

## Results

**Behavior.** Table 2 presents a summary of behavioral measures.

A within-subject two-way ANOVA of the female vs. male classification accuracy yielded no significant effects of facial identity ($F(1,42) = 0.17$, $p = 0.680$, $\eta_G^2 < 0.001$) and makeup ($F(1,42) = 0.02$, $p = 0.898$, $\eta_G^2 < 0.001$) or the interaction between facial identity and makeup ($F(1,42) = 1.84$, $p = 0.182$, $\eta_G^2 = 0.005$). Cohen's *dz* for the makeup factor was 0.020.

A within-subject two-way ANOVA of the female vs. male classification RTs yielded significant effects of facial identity ($F(1,42) = 11.86$, $p = 0.001$, $\eta_G^2 = 0.011$) and makeup ($F(1,42) = 27.85$, $p < 0.001$, $\eta_G^2 = 0.004$) but no significant effect of the interaction between facial identity and makeup ($F(1,42) = 1.07$, $p = 0.306$, $\eta_G^2 < 0.001$). Cohen's *dz* for the makeup factor was 0.805. Participants classified faces as female more quickly in the makeup condition than in the no makeup condition. Furthermore, participants classified their own faces as female more quickly than other females' faces.

A within-subject two-way ANOVA of attractiveness ratings showed significant effects of facial identity ($F(1,42) = 5.81$, $p = 0.020$, $\eta_G^2 = 0.020$) and makeup ($F(1,42) = 91.58$, $p < 0.001$,

**Table 2. Descriptive statistics for behavioral measures in Experiment 2.**

| | accuracy[1] | reaction time[2] | attractiveness rating | SAM valence rating | SAM arousal rating |
|---|---|---|---|---|---|
| one's own face | | | | | |
| makeup | 0.998 (0.006) | 427.69 (56.09) | 5.16 (2.01) | 6.02 (1.12) | 4.26 (1.69) |
| no makeup | 0.996 (0.008) | 433.27 (57.09) | 2.26 (1.76) | 3.88 (1.64) | 3.79 (1.92) |
| another female's face | | | | | |
| makeup | 0.996 (0.008) | 437.42 (57.57) | 5.63 (2.02) | 6.16 (1.31) | 3.56 (1.72) |
| no makeup | 0.997 (0.008) | 447.06 (60.61) | 2.88 (1.99) | 4.51 (1.39) | 3.05 (1.66) |
| main effect of makeup[3] | | | | | |
| makeup | 0.997 (0.006) | 432.55 (55.95) | 5.40 (1.88) | 6.09 (0.97) | 3.91 (1.46) |
| no makeup | 0.997 (0.007) | 440.16 (56.85) | 2.57 (1.61) | 4.20 (1.19) | 3.42 (1.34) |

Descriptive statistics are presented as mean (SD). SAM, Self-Assessment Manikin; RT, Reaction Time.

[1] Accuracy scores are expressed as the proportion of correct trials to total trials.

[2] Median RT per participant in each condition was calculated. The unit for the RT is milliseconds.

[3] For assessing the main effect of makeup, data were pooled across one's own face and another female's face.

$\eta_G{}^2$ = 0.350) but no significant effect of the interaction between facial identity and makeup ($F$ (1,42) = 0.36, $p$ = 0.553, $\eta_G{}^2$ <0.001). Cohen's $dz$ for the makeup factor was 1.459. Participants rated faces with makeup as more attractive than faces without makeup. Additionally, participants rated other females' faces as more attractive than their own faces.

A within-subject two-way ANOVA of the SAM ratings for emotional valence showed a significant main effect of makeup ($F$(1,42) = 76.08, $p$ <0.001, $\eta_G{}^2$ = 0.327). However, no significant effects were found for facial identity ($F$(1,42) = 3.01, $p$ = 0.090, $\eta_G{}^2$ = 0.020) or the interaction between facial identity and makeup ($F$(1,42) = 3.45, $p$ = 0.070, $\eta_G{}^2$ = 0.008). Cohen's $dz$ for the makeup factor was 1.330. Participants rated faces with makeup as more pleasant than faces without makeup.

A within-subject two-way ANOVA of the SAM ratings for emotional arousal identified a significant main effect of facial identity ($F$(1,42) = 7.32, $p$ = 0.010, $\eta_G{}^2$ = 0.042). However, no significant effects were found for makeup ($F$(1,42) = 3.50, $p$ = 0.069, $\eta_G{}^2$ = 0.020) or the interaction between facial identity and makeup ($F$(1,42) = 0.02, $p$ = 0.895, $\eta_G{}^2$ <0.001). Cohen's $dz$ for the makeup factor was 0.285. One's own face was more emotionally arousing than other females' faces. Both faces with makeup and without makeup were found to be mildly arousing.

**Event-related potentials.** During the preprocessing of EEG data, an average of 0.05 channels (SD = 0.21, range: 0–1) was rejected, and an average of 1.86 independent components (SD = 0.71, range: 1–3) was removed as ocular components. None of the 10 channels located in the two ROIs were rejected by any participant. After rejecting bad epochs, not <84% of original trials remained for ERP averaging per participant for each condition (makeup × one's own face: M = 113.49 trials, SD = 3.25 trials; no makeup × one's own face: M = 113.63 trials, SD = 3.18 trials; makeup × another female's face: M = 112.70 trials, SD = 3.45 trials; no makeup × another female's face: M = 112.72 trials, SD = 3.08 trials).

Fig 4 displays the grand average ERP waveforms over four occipitotemporal and six centro-parietal electrodes that were selected a priori for each condition (panel A) and the mean amplitudes of the N170, EPN, P300, and LPP components for each condition (panel B).

A within-subject two-way ANOVA of the mean N170 amplitudes revealed no significant effect of makeup ($F$(1,42) = 2.20, $p$ = 0.145, $\eta_G{}^2$ <0.001), a significant effect of facial identity ($F$(1,42) = 6.68, $p$ = 0.013, $\eta_G{}^2$ = 0.001), and a significant interaction between facial identity and makeup ($F$(1,42) = 5.19, $p$ = 0.028, $\eta_G{}^2$ = 0.001). The analysis of simple main effects confirmed that another female's face with makeup elicited more negative N170 amplitudes than that without makeup ($F$(1,42) = 6.12, $p$ = 0.018, $\eta_G{}^2$ = 0.002). Additionally, viewing one's own face without makeup evoked more negative N170 amplitudes than viewing another female's face without makeup ($F$(1,42) = 13.08, $p$ <0.001, $\eta_G{}^2$ = 0.003). The mean N170 amplitudes for each condition were as follows: makeup × one's own face (M = −0.37 μV, SD = 3.16 μV), no makeup × one's own face (M = −0.44 μV, SD = 3.14 μV), makeup × another female's face (M = −0.36 μV, SD = 3.12 μV), and no makeup × another female's face (M = −0.11 μV, SD = 3.15 μV). The mean N170 amplitudes for the faces with and without makeup pooled across self-face and another female's face were −0.36 μV (SD = 3.12 μV) and −0.27 μV (SD = 3.13 μV), respectively. Cohen's $dz$ for the makeup factor was 0.226.

A within-subject two-way ANOVA of the mean P200 amplitudes revealed a significant main effect of facial identity ($F$(1,42) = 52.27, $p$ <0.001, $\eta_G{}^2$ = 0.017) but no significant main effect of makeup ($F$(1,42) = 0.23, $p$ = 0.636, $\eta_G{}^2$ <0.001) as well as no significant interaction between facial identity and makeup ($F$(1,42) = 0.006, $p$ = 0.941, $\eta_G{}^2$ <0.001). Less positive P200 amplitudes were observed for one's own face than another female's face (makeup × one's own face: M = 3.47 μV, SD = 3.83 μV; no makeup × one's own face: M = 3.43 μV, SD = 3.88 μV; makeup × another female's face: M = 4.50 μV, SD = 4.04 μV; no makeup × another female's face: M = 4.48 μV, SD = 3.99 μV). The mean P200 amplitudes for

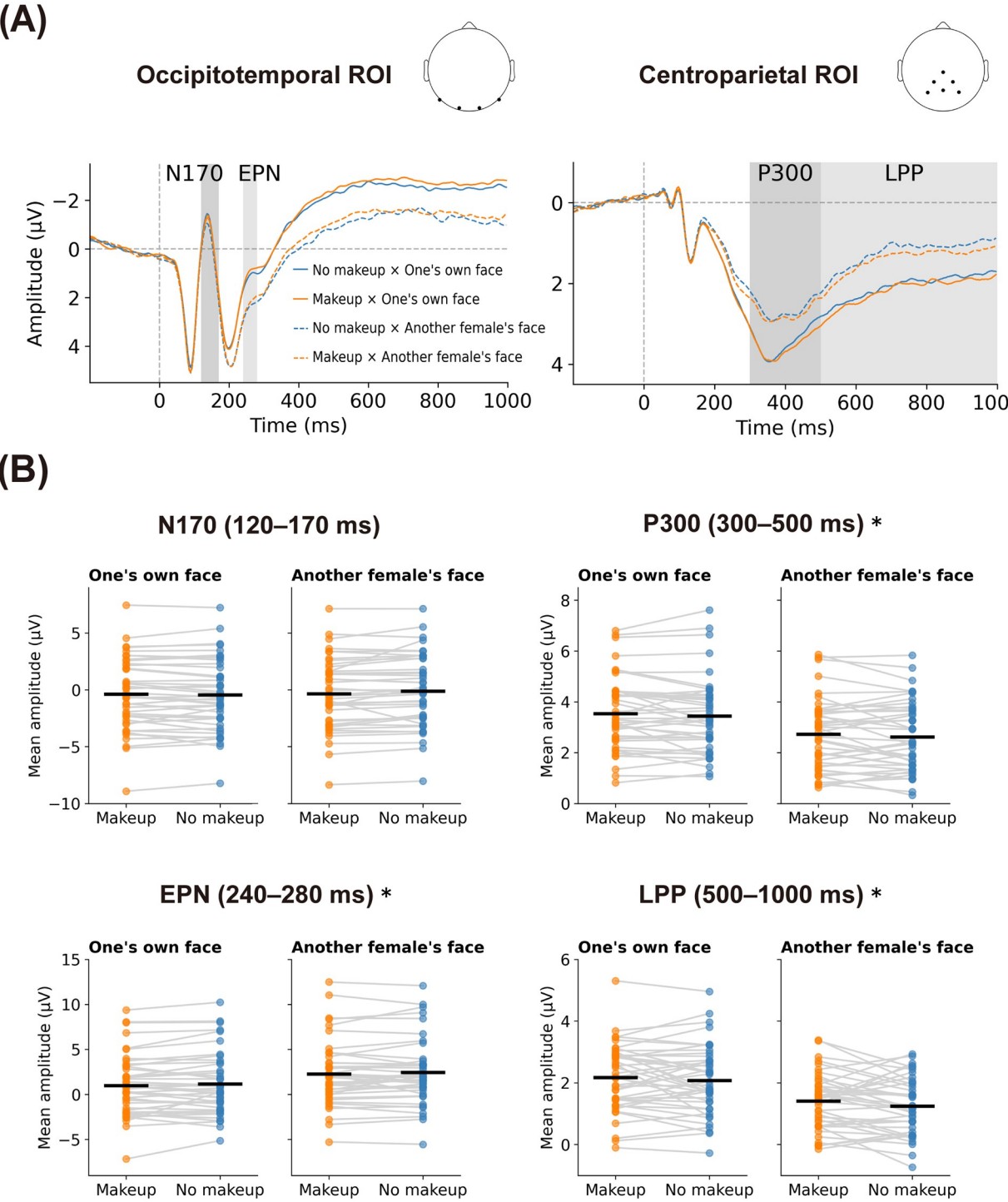

**Fig 4. ERP results in Experiment 2.** Panel A shows the grand average ERP waveforms elicited by facial images in the occipitotemporal and centroparietal ROIs. Panel B shows repeated measures plots visualizing amplitudes of the N170, EPN, P300, and LPP components across the makeup conditions (makeup vs. no makeup), aggregated either by one's own face (left panel) or by another female's face (right panel). Horizontal black lines represent the mean amplitudes for each condition. Asterisks indicate a significant main effect of makeup pooled across face images of different identities determined by a within-subject two-way ANOVA ($p < 0.05$). ERP, event-related potential; ROI, region of interest; EPN, early posterior negativity; LPP, late positive potential; ANOVA, analysis of variance.

the faces with and without makeup pooled across self-face and another female's face were 3.99 μV (SD = 3.89 μV) and 3.95 μV (SD = 3.90 μV), respectively. Cohen's *dz* for the makeup factor was 0.073.

A within-subject two-way ANOVA of the mean EPN amplitudes indicated significant effects of facial identity ($F(1,42)$ = 65.57, $p$ <0.001, $\eta_G^2$ = 0.033) and makeup ($F(1,42)$ = 4.48, $p$ = 0.040, $\eta_G^2$ = 0.001) but no significant interaction between facial identity and makeup ($F(1,42)$ = 0.006, $p$ = 0.938, $\eta_G^2$ <0.001). One's own face evoked more negative EPN amplitudes than another female's face, and faces with makeup evoked more negative EPN amplitudes than faces without makeup (makeup × one's own face: M = 0.99 μV, SD = 3.38 μV; no makeup × one's own face: M = 1.16 μV, SD = 3.47 μV; makeup × another female's face: M = 2.28 μV, SD = 3.60 μV; no makeup × another female's face: M = 2.43 μV, SD = 3.61 μV). The mean EPN amplitudes for the faces with and without makeup pooled across self-face and another female's face were 1.64 μV (SD = 3.43 μV) and 1.79 μV (SD = 3.48 μV), respectively. Cohen's *dz* for the makeup factor was 0.323.

A within-subject two-way ANOVA of the mean P300 amplitudes indicated significant effects of facial identity ($F(1,42)$ = 128.33, $p$ <0.001, $\eta_G^2$ = 0.077) and makeup ($F(1,42)$ = 4.63, $p$ = 0.037, $\eta_G^2$ = 0.001) but no significant interaction between facial identity and makeup ($F(1,42)$ = 0.001, $p$ = 0.973, $\eta_G^2$ <0.001). One's own face evoked more positive P300 amplitudes than another female's face, and faces with makeup evoked more positive P300 amplitudes than faces without makeup (makeup × one's own face: M = 3.54 μV, SD = 1.44 μV; no makeup × one's own face: M = 3.44 μV, SD = 1.49 μV; makeup × another female's face: M = 2.73 μV, SD = 1.40 μV; no makeup × another female's face: M = 2.62 μV, SD = 1.37 μV). The mean P300 amplitudes for the faces with and without makeup pooled across self-face and another female's face were 3.13 μV (SD = 1.39 μV) and 3.03 μV (SD = 1.40 μV), respectively. Cohen's *dz* for the makeup factor was 0.328.

A within-subject two-way ANOVA of the mean LPP amplitudes indicated significant effects of facial identity ($F(1,42)$ = 71.41, $p$ <0.001, $\eta_G^2$ = 0.140) and makeup ($F(1,42)$ = 4.34, $p$ = 0.043, $\eta_G^2$ = 0.004) but no significant interaction between facial identity and makeup ($F(1,42)$ = 0.27, $p$ = 0.605, $\eta_G^2$ <0.001). One's own face evoked more positive LPP amplitudes than another female's face, and faces with makeup evoked more positive LPP amplitudes than faces without makeup (makeup × one's own face: M = 2.18 μV, SD = 1.08 μV; no makeup × one's own face: M = 2.08 μV, SD = 1.11 μV; makeup × another female's face: M = 1.41 μV, SD = 0.91 μV; no makeup × another female's face: M = 1.24 μV, SD = 0.91 μV). The mean LPP amplitudes for the faces with and without makeup pooled across self-face and another female's face were 1.79 μV (SD = 0.94 μV) and 1.66 μV (SD = 0.92 μV), respectively. Cohen's *dz* for the makeup factor was 0.318.

## Discussion

Our primary finding was that seeing makeup both on one's own face and on another female's face led to more negative EPN amplitudes and more positive P300/LPP amplitudes. Thus, Experiment 2 replicated the effects of makeup that were observed for one's own face on the EPN and LPP amplitudes in Experiment 1 and extended these effects to other female faces. These findings are consistent with the findings of existing research that has demonstrated that facial makeup improves the perceived attractiveness of both an individual's own face and others' faces [9–18].

Our study also showed that the occipitotemporal P200 exhibited less positive amplitudes to one's own face than another female's face [59–61]. This finding assures that the participant's brain responses discriminated between their own face and another female's face. The results of

the RTs also demonstrated the advantages of self-face processing. Participants identified their own face faster than another female's face, as reported in other studies [74]. These findings provide further empirical support for the assertion that the effects of makeup on attractiveness-related neural responses are not limited to their own faces.

In contrast with the results of Experiment 1, faces with makeup evoked more positive P300 amplitudes than those without makeup. This finding may reflect the influence of makeup on sexual dimorphism in the human face. Several studies have shown that makeup exaggerates the luminance and color contrasts, which are inherently, on average, higher in women's faces than in men's faces [3, 4]. Consistent with this idea, participants classified faces as female faster when presented with makeup than when presented without makeup, suggesting that makeup helped them identify the faces as female. Based on our findings, as well as previous research, it seems reasonable to assume that participants quickly paid attention to facial regions that were exaggerated by makeup soon after stimulus onset to improve their performance in the female vs. male classification task, resulting in the makeup's enhancement of neural activities reflecting the initial stage of attentional allocation.

Experiment 2 showed interesting results. In particular, more negative N170 amplitudes were elicited when participants were shown other female faces with makeup, while no significant difference in the amplitude of N170 waveforms was registered between the makeup and no makeup condition when participants were shown their own face. This finding was consistent with the findings of Tanaka [28], suggesting that lipstick promotes an attentional focus on the mouth of another female's face. Moreover, viewing one 's own face without makeup elicited more negative N170 amplitudes than viewing another female's face without makeup. These results support Keyes et al.'s [75] finding that participants showed more negative N170 amplitudes for their own faces than for their friends' faces or strangers' faces. The authors argued that one's own face had an advantage in the perceptual processing of face individualization during the N170 time window. However, further research has established that Keyes et al.'s findings could be better explained by the view that their experimental procedure made one's own face activate an analytical processing of individual local facial features more strongly [59–61, 76]. Indeed, research has reported that N170 amplitudes were more negative when facial features, such as eyes, are isolated instead of the whole face [77, 78]. Considering these findings, when encountering another woman's face with makeup, participants might have focused more on a specific facial feature exaggerated by makeup to identify the face as female. Contrastingly it seems reasonable to assume that participants did not have to use such an analytical approach to identify their own face with makeup because they were as familiar with their own face with self-applied makeup as with their own face without makeup.

Although makeup was not explicitly related to gender decisions in Experiment 2, the presence of makeup modulated participants' behavior and attractiveness-related ERP amplitudes. Our findings suggest that facial makeup automatically captures an observer's visual attention because of its motivational significance. This property can be observed for other motivationally salient stimuli, such as erotic or threatening stimuli [43–47]. In summary, these results suggest that facial makeup implicitly modulates our behavior and potentiates its neural correlates.

## General discussion

In a series of two experiments, participants rated faces wearing makeup as more attractive than faces without makeup. Our results offer further corroboration of previous findings, which suggest that makeup increases perceived facial attractiveness [9–18]. The most important finding of our study was that faces with makeup elicited more negative EPN and more positive LPP amplitudes than did faces without makeup in both Experiment 1 and Experiment

2. The effect sizes (within-subject *dz*) of makeup on these ERP amplitudes ranged from 0.3 to 0.4, which can be regarded as small to medium according to Cohen's convention [79]. The EPN and LPP waveforms have been used to index automatic and sustained allocation of visual attention to motivationally salient stimuli, respectively [44, 50–52]. Moreover, previous behavioral findings revealed that people are spontaneously motivated to detect and keep looking at attractive faces [21, 53–55]. Considering these findings, along with the fMRI findings that facial makeup activates the medial orbitofrontal cortex [20], our study suggests that faces with makeup are processed as rewards. The results provide evidence in support of the claim that a positive affective response by viewing faces with makeup yielded more pronounced EPN and LPP responses. Our study is the first to reveal that facial makeup potentiates the EPN and LPP components, reflecting spontaneously motivated visual attention to affective stimuli, including attractive faces.

Alternative explanations for the more pronounced EPN and LPP responses can be provided. First, the results might reflect participants' unpleasant experiences when viewing those pictures because it has been established that both pleasant and unpleasant stimuli induce more negative EPN and more positive LPP amplitudes than emotionally neutral stimuli [43, 45–47, 49, 50, 52]. However, in Experiment 2, SAM results indicated that higher facial attractiveness scores were accompanied by more pleasant emotions. Second, the results might reflect differences in RTs for classifying faces. However, this explanation does not seem plausible because the makeup-induced ERP modulations were constant in both the experiments, whereas the effects of makeup on speed of face classification were different between the two experiments depending on the task demand: makeup slowed down and speeded up participant's responses in Experiment 1 (although not significant) and Experiment 2, respectively. The discrepancy in the RT results can be explained by the relationship between the effects of makeup on facial appearance and the task demands. Makeup reduces facial distinctiveness by obscuring skin roughness and blotches [8–9, 15], and makes faces look more feminine by increasing facial luminance and color contrast [3, 4]. In Experiment 1, the participant had to judge whether a face presented on the screen was her own face with or without makeup. In this task, faces without makeup were possibly easier to be discriminated than the faces with makeup because of the former's greater distinctiveness, even though the RT difference did not reach the statistical significance. In Experiment 2, the participant had to judge whether a presented face was male or female. It is possible that increased facial femininity with makeup helped participants to classify faces as female in this task. Taken together, it is reasonable to assume that the more pronounced EPN and LPP responses for faces with makeup reflect positive affective responses associated with perceived facial attractiveness.

Another finding was that the effects of makeup on the EPN and LPP amplitudes occurred not only in response to other females' faces but also in response to one's own face. These results provide neural evidence that is consistent with behavioral findings regarding the effects of makeup on perceived facial attractiveness [10, 16]. This finding is important because makeup contributes to one's self-image and daily mood [5, 16, 56].

Another key finding was that faces with makeup elicited more negative EPN and more positive LPP amplitudes regardless of whether the presence of makeup was explicitly relevant to the participant's task. This finding was consistent with the finding that makeup had implicit associations with positive evaluative concepts in addition to increasing explicit perceptions of facial attractiveness [19]. Considering these findings, the pleasant feelings triggered by faces with makeup likely act as a reward, the repetition of which has resulted in faces with makeup having become motivationally significant stimuli that can capture visual attention automatically.

In our study, we asked participants to self-apply their makeup. Previous ERP studies used facial images with professionally applied makeup or using a digital makeup simulator [9, 28]. These types of makeup can sometimes appear odd or exaggerated and do not necessarily suit participants' preferred makeup styles [18]. Our study has promising practical implications for the role of facial makeup in daily life.

The findings for N170 and P300 varied across the two experiments. We found the effects of makeup on N170 amplitudes only for another female's face but not for one's own face. Furthermore, makeup that was applied to one's own face did not alter N170 amplitudes, regardless of differences in the tasks between the two experiments. Because there were substantial differences in N170 amplitudes between one's own face and another female's face without makeup, N170 enhancement for another female's face with makeup could be explained by the view that participants employed a strategy focusing on a local facial feature to perform the face classification task efficiently. Contrastingly, facial makeup elicited more positive P300 amplitudes only when it aided quick decision-making in face-classification responses. RT data implied that participants used different strategies for the assignment of faces across the two experiments, differentiating in where they would look at immediately after stimulus onset. To clarify this issue, future studies might need to use an eye tracker. Although more evidence is warranted before reaching definitive conclusions about the effect of makeup on N170 and P300 components, these components presumably reflect a qualitatively different internal mental process from what the EPN and LPP reflect.

Several limitations of our study should be noted. First, we restricted participants' ages to 25–35 years in our study. Our goal was to eliminate any confounding effects of age on the measures we were interested in. However, previous literature has revealed that facial attractiveness is strongly related to age, especially in women. Russell et al. [80] demonstrated that females' faces have greater facial contrasts than males' faces and that facial contrasts decrease as age increases. Moreover, makeup emphasizes youth-related facial contrasts [3, 4, 64]. These studies suggest that the effects of makeup on EPN and LPP amplitudes might be larger for middle-aged and older women than for younger women. Therefore, age differences can be considered in future studies. Furthermore, cultural differences should be examined in future research because the effects of makeup on perceived facial attractiveness have been reported in many countries in addition to Japan [9–18].

Second, we asked all participants to use five types of makeup items and did not control their makeup styles. Mullen et al. [11] showed that eye makeup contributed most strongly to perceived facial attractiveness among female observers. Etcoff and others [13] revealed that different makeup styles are associated with different facial impressions. Furthermore, Tagai et al. [9] suggested that different makeup styles altered the structural encoding of faces differently. Future in-depth research should identify the individual effects of different makeup items while strictly controlling low-level visual features, which can substantially influence early to middle latency ERP components, such as the N170 and EPN. Additionally, future studies should investigate the effect of another style of self-applied makeup because the makeup style when participants would prefer while going out with their friends on a weekend were likely to be just one of the varieties of their makeup styles.

Third, we did not include male participants in our study. Several fMRI and ERP studies have identified that male participants showed stronger brain responses to attractive opposite-sex faces than female participants [23–25, 36, 40]. A research question prompted by these findings is whether male participants elicit more negative EPN and more positive LPP amplitudes in response to females' faces with makeup than female participants. The replication of our study with both male and female participants remains necessary to answer this question.

Finally, an implicit task, where facial makeup is not explicitly related to task demands, different from the one used in our study should be examined. In our study, we manipulated the task relevance of facial makeup. However, researchers can also manipulate the task relevance of attractiveness, as reported by Schacht et al. [39]. Schacht et al. showed that facial attractiveness had larger effects on EPN and LPP amplitudes during attractiveness ratings than during gender classification. Future research should compare the effects of a facial makeup on EPN and LPP amplitudes when participants are performing attractiveness rating tasks and gender classification tasks.

## Conclusions

In summary, our study is the first to demonstrate that facial makeup potentiates EPN and LPP waveforms, which are believed to be sensitive to motivationally salient stimuli. We confirmed that makeup affects the EPN and LPP amplitudes when presented with images of one's own face or other females' faces, regardless of which observer was aware of the presence of makeup. This study provides additional evidence that facial makeup adds visually rewarding value to the physical appearance of human faces.

## Acknowledgments

The authors are grateful to Yuriko Saheki, Fumika Nakagawa, Ayaka Ishii, Mie Sugino, Natsumi Uehara, and Taro Munakata for their assistance in conducting the experiments and to Shuntaro Okazaki, Keiko Tagai, and Keith Kawabata Duncan for their helpful comments on the work and suggestions regarding drafts of this manuscript. We express our appreciation to Fumie Sugiyama, a professional photographer, for her assistance with photography and photo editing. We would like to thank Editage (www.editage.com) for English language editing.

## Author Contributions

**Conceptualization:** Tomohiro Arai, Hiroshi Nittono.

**Data curation:** Tomohiro Arai.

**Formal analysis:** Tomohiro Arai.

**Funding acquisition:** Tomohiro Arai.

**Investigation:** Tomohiro Arai.

**Methodology:** Tomohiro Arai, Hiroshi Nittono.

**Project administration:** Tomohiro Arai.

**Resources:** Tomohiro Arai.

**Supervision:** Hiroshi Nittono.

**Writing – original draft:** Tomohiro Arai.

**Writing – review & editing:** Hiroshi Nittono.

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
