## [Decision Letter · Decision Letter 0]

13 Feb 2022

PONE-D-21-33037Cosmetic makeup enhances facial attractiveness and affective neural responsesPLOS ONE

Dear Dr. Arai,

Thank you for submitting your manuscript to PLOS ONE. After careful consideration, we feel that it has merit but does not fully meet PLOS ONE’s publication criteria as it currently stands. Therefore, we invite you to submit a revised version of the manuscript that addresses the points raised during the review process.

We look forward to receiving your revised manuscript.

Kind regards,

Vilfredo De Pascalis

Academic Editor

PLOS ONE

Journal Requirements:

2. We note that Figures 1 and 3 includes an image of a [patient / participant / in the study].

If you are unable to obtain consent from the subject of the photograph, you will need to remove the figure and any other textual identifying information or case descriptions for this individual

Additional Editor Comments:

Although the study topic is enjoyable for both Reviewers, Reviewer#1 raised critical methodological issues that they must solve. First, the authors should connect the lacking Introduction assumptions and the Discussion. Reviewer#1 suggests that statistical analyses of the behavioral data should be re-performed, and authors should also provide a valuable Discussion of facial discrimination. Reviewerr#2 is more optimistic, although it raises several minor points that the authors should address. I am confident that the authors resolve all submitted comments, making them improve the quality of their paper.

Reviewers' comments:

Reviewer's Responses to Questions

**Comments to the Author**

1. Is the manuscript technically sound, and do the data support the conclusions?

Reviewer #1: Partly

Reviewer #2: Yes

2. Has the statistical analysis been performed appropriately and rigorously? 

Reviewer #1: No

Reviewer #2: Yes

3. Have the authors made all data underlying the findings in their manuscript fully available?

Reviewer #1: Yes

Reviewer #2: Yes

4. Is the manuscript presented in an intelligible fashion and written in standard English?

Reviewer #1: Yes

Reviewer #2: Yes

5. Review Comments to the Author

Reviewer #1: The literature is reviewed in a clear and concise manner. However, the aim of the present study is vague. The authors examined whether makeup evokes greater attractiveness-related brain responses when presented with an individual’s own face than when presented with another person’s face (p. 7, line 151). It is unclear whether their assumptions regarding changes in brain responses depend on makeup effects, facial attractiveness, or discrimination between self and others (p. 9, line 184). The authors should clearly explain the relationship between task processes and ERP components.

In Experiment 1, the authors found that participants identified their own faces without makeup faster than they did with makeup. Participants rated their own made-up faces more attractive than without makeup. Previous studies have indicated that people detect attractive faces automatically and keep looking at them (p. 7, line 139). Are the present results consistent with previous findings? In addition, RTs derived from Experiment 2 were faster for participant’s own made-up faces than for their own faces without makeup. Thus, I wonder whether the behavioral results are reliable. It is well known that RT data do not follow a normal distribution. The authors should perform a logarithmic transformation of RT data (or use median RTs) and conduct the statistical analysis again.

Interpretations of results from Experiment 1 are not consistent. The authors stated that when participants viewed their own faces with makeup, larger EPN and LPP waveforms were elicited than when they viewed their own faces without makeup (p. 18, line 400). However, although the amplitude of the LPP component was greater with makeup, the amplitude of the EPN component was greater without makeup (Fig. 2). Do the amplitudes of ERP components reflect differences in facial attractiveness or RTs?

For Experiment 2, why did the authors analyze the P200 component? The Introduction offers no explanation of mental processes based on P200. It is very confusing. If the P200 component is important in recognizing one’s face, it should be analyzed in Experiment 1 as well.

The authors used different EEG setups in Experiments 1 and 2. Locations of electrodes also differed between the two experiments. Thus, the results of Experiment 1 are not directly comparable to those of Experiment 2. The authors should explain the reasons for these different EEG arrangements.

The authors reported significant differences in brain responses between experimental conditions, but the effect size in each case was very small. I am not sure that these results are robust. Are there any criteria regarding the effect size?

My impression is that the manuscript lacks logical connections between the Introduction and the Discussion. First, what is the ecological relevance of distinguishing made-up faces of self from those of others? In daily life, we see other's faces, but we rarely see our own. I think that makeup can change one’s mood (or increase extroversion along with self-confidence). Do such changes in mood affect brain responses? Makeup can also enhance the salience of facial features (luminance, color, contrast, etc.). The authors discuss effects of makeup on facial attractiveness and visual attention, but not the effect of salience on object detection. They should clearly explain how ERP components are associated with mental processes of face recognition.

I cannot read the descriptions in the figures (e.g., Figs 2 and 4) because of low resolution. Please improve the quality of figures.

Reviewer #2: On the whole, this study has strengths that are worth mentioning -- I have a few clarification points that I would like to see, however.

First of all, the literature review is thorough, and the novelty of the research is clear in terms of how it expands on the literature. I appreciated the expansion of the research to focus on reactions to one's own face, to comparing one's own face with another's, and the exploration of whether the results would still occur under implicit conditions. Likewise, having the more realistic notion of makeup that has been self-applied has its benefits, although it also provided some downsides, which I'll expand on below. The description of the methodology used was thorough, and the sample size was sufficient for research of this type, and for the most part, I found the analysis to likewise be complete and comprehensible. The discussion was clear at highlighting the key findings, and was reasonably transparent in terms of methodological challenges (i.e having them apply their own makeup without strict consistency).

There were a few things I would like to see added to the study:

1.) The description of the makeup the participants were told to apply was 'makeup that they would normally wear while going out with their friends on the weekend.' (p. 12, ln. 251-252, also in Study 2 on p. 22, ln 491-492). On page 19, ln. 413, the makeup worn is classified as 'ordinary makeup'. This is not day-to-day wear -- it is makeup explicitly worn only on the weekends for a night out. This, to a certain extent, could help explain some of the results in terms of reaction time, as participants would be more used to seeing their bare faces, or the makeup they may have worn while at work, as opposed to the rarer, dressier makeup they were asked to put on. The researchers should further clarify how the type of makeup could differ from daily wear, and not refer to it as 'ordinary' makeup, as that could be confusing.

2.) While the effect sizes were appropriate to the type of test used, the authors should acknowledge effect size in their discussion of results, making sure to specify not only a significant finding, but clarifying when a result was a large or more modest effect to better contextualize the results -- as well as briefly identifying what is a typical effect size for studies using this methodology.

3.) The authors report all cell means and standard deviations, but in Study 2, there are several main effect tests that are significant; providing the main effect means and standard deviations would be preferable in addition to cell means. (p.27, lns 572, 573, 579, 589 -- or added to Table 2; p. 29 ln 627 and 635; p. 30 ln 644, 651, 652)

6. PLOS authors have the option to publish the peer review history of their article (what does this mean?). If published, this will include your full peer review and any attached files.

Reviewer #1: **Yes: **Hirohito M. Kondo

Reviewer #2: **Yes: **Caitlin A. J. Powell

---

## [Author Response · Author response to Decision Letter 0]

24 Apr 2022

We thank the editor and reviewers for positively receiving our manuscript. We are grateful to receive constructive and useful suggestions. In the revised manuscript that we submit with this response letter, we have undertaken required changes to address all the comments made. All the changes in the revised manuscript (in reference to original submitted manuscript) are in red text in the "Revised Article with Changes Highlighted” file. You can also find our point-by-point response to each comment in the “Response to reviewers” file.

---

## [Decision Letter · Decision Letter 1]

7 Jul 2022

PONE-D-21-33037R1Cosmetic makeup enhances facial attractiveness and affective neural responsesPLOS ONE

Dear Dr. Arai,

Thank you for submitting your manuscript to PLOS ONE. After careful consideration, we feel that it has merit but does not fully meet PLOS ONE’s publication criteria as it currently stands. Therefore, we invite you to submit a revised version of the manuscript that addresses the points raised during the review process.

I am very sorry for the long delay. I have found it seriously difficult to find expert reviewers.

However, I think the paper still needs a minor revision at this stage.

Given the long delay, I suggest the authors revise their manuscript by solving the reviewer comments and then submit their last revision for acceptance.

Please accompany the revised manuscript with a detailed letter explaining how you have responded to each of the reviewers' points, and where the changes appear in the revised manuscript.

We look forward to receiving your revised manuscript.

Kind regards,

Vilfredo De Pascalis

Academic Editor

PLOS ONE

Journal Requirements:

Additional Editor Comments (if provided):

I am very sorry for the long delay. I have found it seriously difficult to find expert reviewers.

However, I think the paper still needs a minor revision at this stage.

Given the long delay, I suggest the authors revise their manuscript by solving the reviewer comments and then submit their last revision for acceptance.

Please accompany the revised manuscript with a detailed letter explaining how you have responded to each of the reviewers' points, and where the changes appear in the revised manuscript.

Reviewers' comments:

Reviewer's Responses to Questions

**Comments to the Author**

1. If the authors have adequately addressed your comments raised in a previous round of review and you feel that this manuscript is now acceptable for publication, you may indicate that here to bypass the “Comments to the Author” section, enter your conflict of interest statement in the “Confidential to Editor” section, and submit your "Accept" recommendation.

Reviewer #1: (No Response)

2. Is the manuscript technically sound, and do the data support the conclusions?

Reviewer #1: Yes

3. Has the statistical analysis been performed appropriately and rigorously? 

Reviewer #1: Yes

4. Have the authors made all data underlying the findings in their manuscript fully available?

Reviewer #1: Yes

5. Is the manuscript presented in an intelligible fashion and written in standard English?

Reviewer #1: Yes

6. Review Comments to the Author

Reviewer #1: The several descriptions remain difficult to understand. I pointed them out in my previous comments, but the authors did not update their manuscript. For example, in lines 397-399, the authors stated that “Viewing one’s own face with makeup on resulted in larger EPN waveforms than viewing one’s own face without makeup (makeup: M = 2.60 μV, SD = 2.62 μV; no makeup: M = 2.92 μV, SD = 2.44 μV).” What does a larger waveform mean? That statement does not depend on whether the waveform indicates positive or negative deflection. The authors also stated that “Another key finding was that faces with makeup elicited larger EPN and LPP amplitudes regardless of whether the presence of makeup was explicitly relevant to the participant’s task" (lines 781-783). This is misleading. As described above, the amplitude of the EPN component was larger for the non-makeup condition. I found that the EPN component was negatively biased for the makeup condition than for non-makeup condition, whereas the LPP component was positively biased for the makeup condition than for non-makeup condition. The authors should check their statements in all the sections and reword them appropriately.

Technical comment: the authors should enlarge the font size in the figures.

7. PLOS authors have the option to publish the peer review history of their article (what does this mean?). If published, this will include your full peer review and any attached files.

Reviewer #1: **Yes: **Hirohito M. Kondo

---

## [Author Response · Author response to Decision Letter 1]

18 Jul 2022

We thank the editor and reviewers for positively receiving our manuscript. We are grateful to receive constructive and useful suggestions. The manuscript has been rechecked and the necessary changes have been made in accordance with the reviewers’ suggestions. Thank you for your consideration. I look forward to hearing from you.

---

## [Editor Report · Decision Letter 2]

29 Jul 2022

Cosmetic makeup enhances facial attractiveness and affective neural responses

PONE-D-21-33037R2

Dear Dr. Arai,

We’re pleased to inform you that your manuscript has been judged scientifically suitable for publication and will be formally accepted for publication once it meets all outstanding technical requirements.

Kind regards,

Vilfredo De Pascalis

Academic Editor

PLOS ONE

Additional Editor Comments (optional):

First of all, I am sorry for the long-lasting time required to revise this manuscript. I have had difficulties finding expert reviewers serving for this manuscript and thank the authors for their patience.

I see that the current revised version is quite improved after addressing the insightful reviewer' suggestions. The authors have addressed all the necessary changes following the reviewers’ suggestions. Thus, I think the manuscript can be accepted for publication at this stage.
---

## [Editor Report · Acceptance letter]

5 Aug 2022

PONE-D-21-33037R2 

Cosmetic makeup enhances facial attractiveness and affective neural responses 

Dear Dr. Arai:

I'm pleased to inform you that your manuscript has been deemed suitable for publication in PLOS ONE. Congratulations! Your manuscript is now with our production department. 

Kind regards, 

on behalf of

Prof. Vilfredo De Pascalis 

Academic Editor

PLOS ONE